# Effect of Detergents Based on Sodium Dodecyl Sulfate on Functional Metrics of Frozen–Thawed Collared Peccary (*Pecari tajacu*) Semen

**DOI:** 10.3390/ani13030451

**Published:** 2023-01-28

**Authors:** Samara S. J. Moreira, Andréia M. Silva, Ana G. Pereira, Romário P. Santos, Maiko R. T. Dantas, João B. F. Souza-Júnior, Paola P. N. Snoeck, Alexandre R. Silva

**Affiliations:** 1Laboratory of Animal Germplasm Conservation, Department of Animal Sciences, Federal University of Semiarid Region–UFERSA, Mossoró 59625-900, Brazil; 2Laboratory of Animal Reproduction, State University of Santa Cruz–UESC, Ilhéus 45662-900, Brazil

**Keywords:** *Tayassuids*, biobanking, sperm, membrane integrity, SDS

## Abstract

**Simple Summary:**

Constant threats to wildlife have led countless researchers to develop or improve conservation strategies, among which the formation of biobanks stands out. The systematic storage of male gametes in freezing temperatures is useful for future use in assisted reproduction and for improving the reproductive management of rare species. In this study, the effect of including Equex STM^®^ paste or different concentrations of sodium dodecyl sulfate (SDS) on various functional metrics of frozen and thawed peccary semen, such as sperm kinetic parameters, membrane functionality and integrity, mitochondrial activity and binding capacity, was verified. In parallel, sperm longevity was evaluated through a thermal resistance test. In general, it was demonstrated that the addition of 0.1% SDS to the Tris–egg yolk–glycerol diluent favored the maintenance of the sperm kinetic parameters of peccaries during freezing and thawing procedures.

**Abstract:**

We evaluated the effects of detergents based on sodium dodecyl sulfoxide (SDS) on the functional parameters of collared peccary frozen–thawed sperm. Semen aliquots from ten individuals were diluted in a Tris–egg yolk–glycerol extender alone or with 0.5% Equex STM^®^ paste or SDS (at 0.1%, 0.3% or 0.5% (*v/v*) concentration). Samples were fast frozen in liquid nitrogen with a post-thaw evaluation of motility, membrane functionality and integrity, mitochondrial activity, sperm binding ability and thermal resistance. The treatments without SDS (41.8 ± 3.5%) and those containing Equex (41.8 ± 4.4%) or 0.1% SDS (41.2 ± 5.5%) provided greater sperm motility (*p* < 0.05) than those containing SDS 0.3% (30.5 ± 4.7%) and 0.5% (31.2 ± 6.3%). Immediately after thawing, only treatments containing 0.1% SDS effectively preserved sperm straightness (STR) when compared to the negative control. All treatments preserved the amplitude of lateral head (ALH) and straightness (STR) during a thermal resistance test (*p* > 0.05), but SDS 0.5% impaired the membrane functionality and mitochondrial activity after thawing (*p* < 0.05). All treatments provided a similar recovery of sperm binding ability after thawing (*p* < 0.05). Our results showed that the addition of 0.1% SDS to the Tris–yolk–glycerol extender optimized the freeze–thaw recovery of peccary semen.

## 1. Introduction

In view of constant threats to the survival of various wild species, the need for efficient conservation strategies has been imminent. Among such strategies, biobanks emerge as important repositories of valuable genetic material from endangered animals [1]. Important studies have focused on the cryopreservation of the male gametes of ecologically important individuals, such as collared peccaries (*Pecari tajacu* Linnaeus, 1758). These peculiar ungulates that originally lived from North to South America have today suffered a decline in their population, having even disappeared in some regions such as eastern and southern Argentina [2].

Since 2010, the protocol for the cryopreservation of collared peccary semen was developed [3] from the incorporation of egg yolk and glycerol with diluents based on Tris [4] or coconut water [5]. One of the great challenges to improving the protocol lies in the short-term peccary sperm survival after thawing [6]. To overcome this obstacle, a preliminary study demonstrated that the addition of Equex STM paste (Nova Chemical Sales, Scituate, MA, USA) would promote a slight extension of just 30 min in the maintenance of sperm longevity after thawing [7]. However, the effect of the paste on other sperm parameters, mainly related to sperm function, still needs to be elucidated. Furthermore, the exact composition of the paste in terms of excipients and other compounds is not known, even if manufacturers highlight sodium dodecyl sulfate (SDS) as the main active component, albeit at an unknown concentration. In general, there is a lack of information regarding the incorporation of isolated SDS with diluents for semen conservation, since most studies have reported the use of commercial pastes containing this substance.

The addition of commercial SDS-based supplements to extenders for semen cryopreservation has contributed not only to increased sperm longevity in different species, but also stands out for maintaining the post-thawing sperm kinetic rate, as recently reported for buffaloes [8]. In addition, the incorporation of SDS-based pastes with diluents has also promoted a positive effect on sperm binding and fertilizing ability in dogs [9], cats [10] and boars [11]. Despite the extensive use of SDS-based supplements in semen freezing in different species, their effects on the mitochondrial activity of sperm from any species remain to be elucidated. This would be important information to show, as Bezerra et al. [7] demonstrated through an ultrastructural analysis using transmission electron microscopy, that cryopreservation promotes mitochondrial vacuolation, thus, possibly contributing to a reduction in post-thawing sperm survival in peccaries.

Therefore, the aim of the present study was to verify the effect of including Equex STM^®^ paste or different concentrations of sodium dodecyl sulfate (SDS) on various functional metrics of frozen–thawed peccary semen such as sperm kinetic parameters, membrane functionality and integrity, mitochondrial activity and sperm binding ability, in addition to evaluating sperm longevity through a thermal resistance test.

## 2. Materials and Methods

### 2.1. Animal Ethics and Husbandry

All experimental procedures were authorized by the ethics committee of the Universidade Federal Rural do Semi-Árido (UFERSA) (nº 05/2020). The protocols were authorized by the Chico Mendes Institute for Biodiversity (opinion nº 37329/3). Sigma Chemical Co. (St. Louis, MO, USA) provided all reagents used in the experiment, unless otherwise specified.

Ten sexually mature collared peccaries, with a mean age of 40 months, were used. The animals were exposed to a natural photoperiod of 12 h and up to three animals were allocated into paddocks (20 m × 3 m), presenting a covered area (3 m × 3 m) at the Wild Animal Multiplication Center located on the UFERSA campus, Mossoró, Brazil (5°10′ S–37°10′ W; medium temperature range, 27–29 °C). For the study, an isocaloric (3300 kcal/kg) and isoprotein (14% protein) diet, supplemented with tropical fruits and water ad libitum, was offered to the animals.

### 2.2. Experimental Design

After obtaining ejaculate from ten individuals, one aliquot of fresh semen was immediately evaluated, and five other aliquots were diluted in Tris–egg yolk–glycerol extenders and used for cryopreservation. Regarding the dilution, the first aliquot received no detergent supplementation (negative control group); the second aliquot was supplemented with 0.5% Equex STM paste (Nova Chemical Sales, Inc., Scituate, MA, USA), constituting a positive control group [7]; the other three semen aliquots were supplemented with SDS only in increasing concentrations of 0.1%, 0.3% or 0.5% *v/v*, as previously tested for boars [12], the domestic species most closely related to peccaries. Fresh and frozen–thawed samples were evaluated for sperm kinetic parameters, membrane functionality and integrity, mitochondrial activity, sperm binding ability and longevity.

### 2.3. Semen Collection and Initial Evaluation

After a 12 h fast, the peccaries were restrained with a net and anesthetized with intravenous propofol (Propovan^®^, Cristália, Fortaleza, Brazil) in bolus (5 mg/kg). During the anesthesia, the cardiac and respiratory parameters of the animals were evaluated [13]. The animals were placed in lateral recumbency, and the semen was obtained through the electroejaculation protocol, as previously established for the species [3]. For this purpose, a portable device (Autojac^®^, Neovet, Campinas, SP, Brazil) connected to a 12 V source provided a stimulatory cycle comprised of 10 stimuli in each voltage, starting from 5 V, followed by a voltage increase in steps from 1 V to 12 V. Each electrical stimulus lasted 3 s, with intermittent breaks of 2 s. The stimulus cycle lasted 10 min from the commencement of the procedure. The electroejaculator probe measured 15 × 1.3 cm, of which 12 cm was introduced into the rectum. Ejaculates were stored in plastic recipients and immediately analyzed for semen volume, which was measured with repeated pipetting. The pH was defined using pH indicator strips (Neutralit^®^, Merck, Bucharest, Romania) [13] and the sperm concentration (in millions of sperm/mL) was determined by counting cells in a Neubauer chamber after dilution (1:2) in a formalized buffered solution (10%) [14].

### 2.4. Semen Freezing–Thawing Procedures

For freezing, the samples were diluted in Tris containing egg yolk (20%), glycerol (3%) and gentamicin (70 μg/mL) [15], and separated according to the following treatments: (A) negative control (no detergent), (B) positive control (0.5% Equex STM^®^ paste (Equex–Nova Chemical Sales, Inc., MA, USA), (C) 0.1% SDS, (D) 0.3% SDS and (E) 0.5% SDS. Semen was frozen according to a fast-freezing curve previously described by Silva et al. [16], in which diluted semen cooled at 15 °C for 40 min in isothermal containers and stabilized at 5 °C for another 30 min in an incubator (Quimis, Diadema, SP, Brazil). Then, samples were packed in 0.25 mL plastic straws that were exposed to nitrogen vapor (5 cm) for 5 min and, finally, stored in a cryobiological cylinder at −196 °C. After 1 week, the samples were thawed in a water bath at 37 °C for 1 min [17], and the functional sperm parameters were immediately evaluated. For a thermal resistance test, samples were incubated in a water bath at 37 °C and re-evaluated at 30 and 60 min.

### 2.5. Computer-Aided Semen Analysis

Information on fresh and frozen–thawed sperm motility characteristics and kinetic parameters was obtained with a computerized semen analysis (IVOS 7.4 G; Hamilton-Thorne Research, Beverly, MA, USA) with pre-established settings for the species, such as a temperature of 37 °C; 60 frames/s; minimum contrast 45; straightness threshold 30%; low-velocity average pathway (VAP) cutoff, 10 m/s; and a medium VAP cutoff of 30 m/s [17]. The parameters evaluated were the total and progressive motility (%), velocity average pathway (VAP; µm/s), velocity straight line (VSL; µm/s), curvilinear velocity (VCL; µm/s), amplitude of lateral head (ALH; µm), beat cross frequency (BCF; Hz), straightness (STR; %) and linearity (LIN; %). When there was a low VAP cutoff (LVC) and medium VAP cutoff (MVC), the overall sperm population was subdivided into four categories: rapid, with VAP > MVC; medium, with LVC < VAP < MVC; slow, with VAP < LVC; and static for the absence of cell motility. For a reliable assessment of sperm motility patterns, the Edit IVOS 7.4 G System Tracks option was used to exclude debris derived from the diluents. There was further dilution in a saline solution (1:2) only when necessary [17].

### 2.6. Sperm Membrane Functionality

A hypoosmotic test using distilled water as the hypoosmotic solution (0 mOsm/L) was used to assess the sperm membrane functionality. Fresh and frozen–thawed semen aliquots were diluted following the proportion of 1:9 and incubated at 37 °C for 45 min. Then, a total of 200 cells were counted, and those with curled tails were judged as having a functional membrane under phase contrast microscopy (Alttion^®^, Wuzhou, China) [18].

### 2.7. Plasma Membrane Integrity and Mitochondrial Activity

The plasma membrane integrity and mitochondrial activity in fresh and frozen–thawed samples were examined simultaneously through the combined use of fluorophores Hoechst 342 (H342; Sigma-Aldrich, St. Louis, MO, USA), Mito Tracker red^®^ (CMXRos, Molecular Probes, M-7512) and propidium iodide (IP, Sigma-Aldrich, Co., St. Louis, MO, USA). A total of 200 cells were counted under epifluorescence microscopy (EFA fluorescent accessory Halogen Lamp Set. Leica. Kista, Sweden). Cells that had a head marked in blue (H-342) were classified as having an intact membrane, while those that had a head marked in red (PI) were identified as having a nonintact membrane. In addition, those with the intermediate piece marked in red were considered to have an active mitochondrial function [17,19].

### 2.8. Sperm Binding Ability Assay

For fresh and frozen–thawed samples, an interaction test with a hen’s egg perivitelline membrane was used to evaluate the sperm binding capacity, as previously established for collared peccaries [20]. Briefly, fresh and infertile eggs, whose membranes were obtained by separating the yolk and white, were used, standardizing cuts of 1 cm^2^, with two membranes per treatment. The semen samples were diluted (1:1) in an incubation solution (114 mM NaCl; 3.1 mM KCl; 0.4 mM NaH2PO4; 10 mM calcium lactate; 25 mM NaHCO3; 10 μg/mL phenol; 1.4 mM caffeine; 2.0 mM CaCl_2_.2H_2_O; 0.5 mM MgCl_2_; 10 mM Hepes; 6 mg/mL BSA; 5.5 mM glucose; 0.45 mM sodium pyruvate; 40 μg/ mL gentamicin; pH 7, 47, 8) and further centrifuged at 700× *g* for 10 min to establish a sperm concentration of 1 × 10^6^ sperm/mL. The membranes, together with the sperm solution, were incubated in a water bath at 38.5 °C for 20 min, washed in 100 μL drops of the incubation solution and kept in Hoechst 33,250 for 15 min. Then, the membranes were stretched over a slide and coverslip and six random fields were analyzed under epifluorescence microscopy (Episcopic Fluorescent attachment EFA Halogen Lamp Set. Leica. Kista, Sweden) for the number of binding spermatozoa.

### 2.9. Statistical Analysis

Data obtained were expressed as mean ± standard error and all analyses were performed using the Statistical Analysis Software, version 8.0 (SAS Institute Inc., Cary, NC, USA). The Shapiro–Wilk test verified the normality of residual and the Levene test evaluated the homogeneity of variance. A two-way ANOVA, which considered the effects of the treatment (SDS-based extenders), incubation time and its interaction, was performed. Then, to assess the potential differences among the means, Tukey’s post hoc test was performed. All pairwise comparisons were considered significant when a *p* < 0.05.

## 3. Results

### 3.1. Fresh Semen Parameters

The peccary ejaculates showed a milky aspect, with a whitish color and pH 7.2 ± 0.1. The mean volume obtained was 3.9 ± 0.4 mL, with a sperm concentration of 470 ± 33.7 × 10^6^ sperm/mL, among which 87.8 ± 4.4% showed motility with kinetic parameters, shown in Table 1. Regarding sperm subpopulations, fresh samples presented 75.7 ± 4.6% rapid, 12.1 ± 2.0% medium, 5.6 ± 2.5% low and 6.7 ± 2.9 static sperm.

Additionally, fresh samples presented an 85.9 ± 3.3% functional membrane, 87.8 ± 2.2% viable sperm and 87.4 ± 2.1% with mitochondrial activity. An average of 204.2 ± 11.6 peccary sperm was bound to the hen eggs’ perivitelline membrane.

### 3.2. Sperm Motility Characteristics, Kinetic Parameters and Subpopulations

Immediately after thawing, the highest concentrations of SDS, 0.3% (30.5 ± 4.7%) and 0.5% (31.2 ± 6.3%), impaired the preservation of sperm motility (*p* < 0.05) when compared to the control treatment without SDS (41.8 ± 3.5%) or to the treatments containing Equex (41.8 ± 4.4%) or 0.1% SDS (41.2 ± 5.5%), as observed in Figure 1. Additionally, the treatment containing 0.1% SDS optimized the preservation of STR immediately after thawing in comparison to the negative control (*p* < 0.05). During the thermal resistance test, all treatments maintained the ALH and STR for 30 min after thawing (*p* < 0.05), while the other kinetic parameters suffered a significant drop at each evaluation at 30 and 60 min.

For sperm subpopulations (Figure 2), the 0.5% SDS increased (*p* < 0.05) the proportion of static sperm in comparison to the negative control and groups containing Equex or SDS 0.1%. This was even more evident during the thermal resistance test, since a significant increase in the number of static spermatozoa was seen in all treatments at each evaluation, with a consequent reduction (*p* < 0.05) in motile spermatozoa in the rapid, medium or slow subpopulations.

### 3.3. Sperm Membrane Functionality and Integrity and Mitochondrial Activity

Immediately after thawing, the highest concentration of SDS (0.5%) impaired the membrane functionality and mitochondrial activity (Figure 3) when compared to other treatments (*p* < 0.05). As for the thermal resistance test, all treatments maintained membrane functionality for 30 min, only suffering a significant drop at 60 min (*p* < 0.05). The membrane integrity and mitochondrial activity parameters declined after 30 min in all treatments (*p* < 0.05). The group containing 0.5% SDS caused a significantly more pronounced decline in membrane functionality and mitochondrial activity over time (*p* < 0.5) compared to the control group.

### 3.4. Sperm-Binding Ability

There were no differences among treatments after thawing with regard to sperm binding ability (Table 2).

## 4. Discussion

In an attempt to contribute to the formation of biobanks, we provided effective information about the actions of SDS-based detergents on some functional parameters of collared peccary sperm submitted to freezing and thawing processes. In general, we found that SDS alone had a positive effect on sperm kinetic parameters, especially regarding motility and straightness (STR). However, an increase in its concentration could cause significant damage to the functional parameters of collared peccary sperm, especially the plasmatic membrane functionality and mitochondrial activity. From an experimental point of view, the possibility of using isolated SDS in the formulation of extenders for semen would be an advantage over the commercial detergent Equex STM paste, since its composition is not exactly known.

SDS is an anionic detergent that seems to potentialize the cryoprotective effect of the egg yolk, allowing for a greater distribution of phospholipid molecules in the yolk and, consequently, providing greater protection to the sperm membrane [21], thus, enhancing other sperm functional parameters, such as the motility. As observed in domestic pigs, in which 0.1% SDS showed a positive effect on sperm motility and a negative effect at concentrations above 0.2% [12], the best results in the present study were achieved when 0.1% SDS was incorporated into the extender for the cryopreservation of collared peccary semen. This SDS concentration promoted not only obtaining adequate values of peccary sperm motility immediately after thawing, but also helped contribute to the maintenance of some kinetic parameters, such as the STR. It is known that some kinetic parameters are jointly considered as indicators of sperm vigor, and their evaluation is used as a criterion for the occurrence of hyperactivation, since such parameters are usually intensified in the capacitation event [22,23]. In this sense, SDS seemed to influence the prevention of the early capacitation of collared peccary spermatozoa from being able to maintain such parameters during a short-time incubation after thawing.

Generally, the sperm plasma membrane is the main target to be damaged by cryopreservation and thawing procedures. This probably occurs due to thermal stress that usually promotes changes in the structure of the phospholipid–protein bilayer [24]. In parallel to thermal and osmotic stress, sperm cells are very sensitive to oxidative stress resulting from freezing and thawing procedures, especially because they have a high concentration of polyunsaturated fatty acids in the membrane and a low percentage of antioxidant agents [25]. Protecting the integrity of the sperm plasma membrane is a crucial point in the development of any sperm cryopreservation protocol. If, on the one hand, the addition of SDS in adequate concentrations can promote the protection of the plasma membrane by increasing the solubility of most egg yolk particles, making them more accessible to spermatozoa, and, thus, improving their freezability, when SDS is used in high concentration in the diluent, free SDS molecules increase and can bind directly to the sperm, with devastating results on the sperm membrane [26]. These negative effects are generally related to the excessive fluidity of the plasmatic membrane of spermatozoa exposed to high SDS concentrations [27], as verified for peccary semen exposed to an extender containing 5% SDS. Interestingly, the harmful effect of the high concentration of SDS (0.5%) on collared peccary spermatozoa was more evident on membrane functionality than on its integrity. In fact, a similar event had previously been described in elephants, in which it was suggested that detergents such as Equex may protect the structural integrity of the membrane, but were not able to maintain its capacity for osmotic regulation [28].

In parallel, the highest concentration of SDS (0.5%) promoted a negative effect on the mitochondrial activity of collared peccary sperm, emphasizing that the detergent may perhaps act on the function of this organelle. In fact, the influence of detergent incorporation on this parameter has been little studied and remains to be better elucidated. In a single study found in the consulted literature, the effect of different diluents on the mitochondrial activity of spermatozoa from Iberian pigs was evaluated, but a direct action of detergents on this parameter was not identified [29].

The action of SDS-based detergents on sperm longevity seems to vary according to the species, since the addition of Equex paste to the freezing extender negatively affected the post-thawing quality of Andalusian donkey spermatozoa [30] and seems not necessarily essential for the cryopreservation of dromedary camel semen [31]. Unlike what is observed for canine spermatozoa, which have a post-thawing longevity of approximately seven hours when exposed to diluents containing the SDS detergent [9], collared peccary spermatozoa began a significant decline in most of their functional parameters as early as 30 min after thawing. Despite this fact being previously known [6], the present study showed that not even the incorporation of detergents based on SDS had the capacity to prolong the sperm longevity of collared peccaries after thawing. This emphasizes the importance of using semen immediately after thawing, associated with an efficient method of monitoring the estrous cycle of females [32], when implementing artificial insemination programs in collared peccaries [33].

In the present study, the incorporation of Equex paste or SDS in different concentrations with the Tris–yolk–glycerol diluent did not influence the binding capacity of collared peccary spermatozoa, since similar results were obtained when these supplements were not present in the diluent. Based on the data obtained of our results, SDS had no toxic effect of on sperm binding phenomena. Moreover, we believe that the use of SDS alone would be a more standardized process than the addition of Equex paste, as its composition is not completely known. However, the assay’s limitations were evident, and only in vivo insemination tests could reveal the real fertilizing capacity of the frozen/thawed semen samples in the absence or presence of SDS, as recently postulated for canine semen [25].

When establishing an efficient diluent for freezing the semen of a given species, it is essential to consider the practical factors for its use. Extenders containing surfactants such as SDS are known to reduce the occurrence of sperm agglutination in thawed brown bear samples [34]. Likewise, less evidence of sperm agglutination was seen in collared peccary spermatozoa in those groups containing SDS-derived detergents, compared to the group that did not. More than that, the solubilization effect promoted by the detergents also seemed to allow for a better visualization of the collared peccary semen samples when they were evaluated with CASA, presumably by dispersing egg yolk–lipid conglomerates formed after the dilution of the semen [35].

## 5. Conclusions

In summary, this study revealed important interactions between the SDS-based detergent and the functional parameters of collared peccary spermatozoa, evidencing a relationship with the proportion of this substance in the diluent, since the use of the highest concentration (0.5%) mainly affected its sperm motility, membrane functionality and mitochondrial activity. In this sense, the incorporation of SDS 0.1% in the Tris–yolk–glycerol diluent optimized the sperm functional parameters of peccaries during freezing and thawing procedures.

## Figures and Tables

**Figure 1 animals-13-00451-f001:**
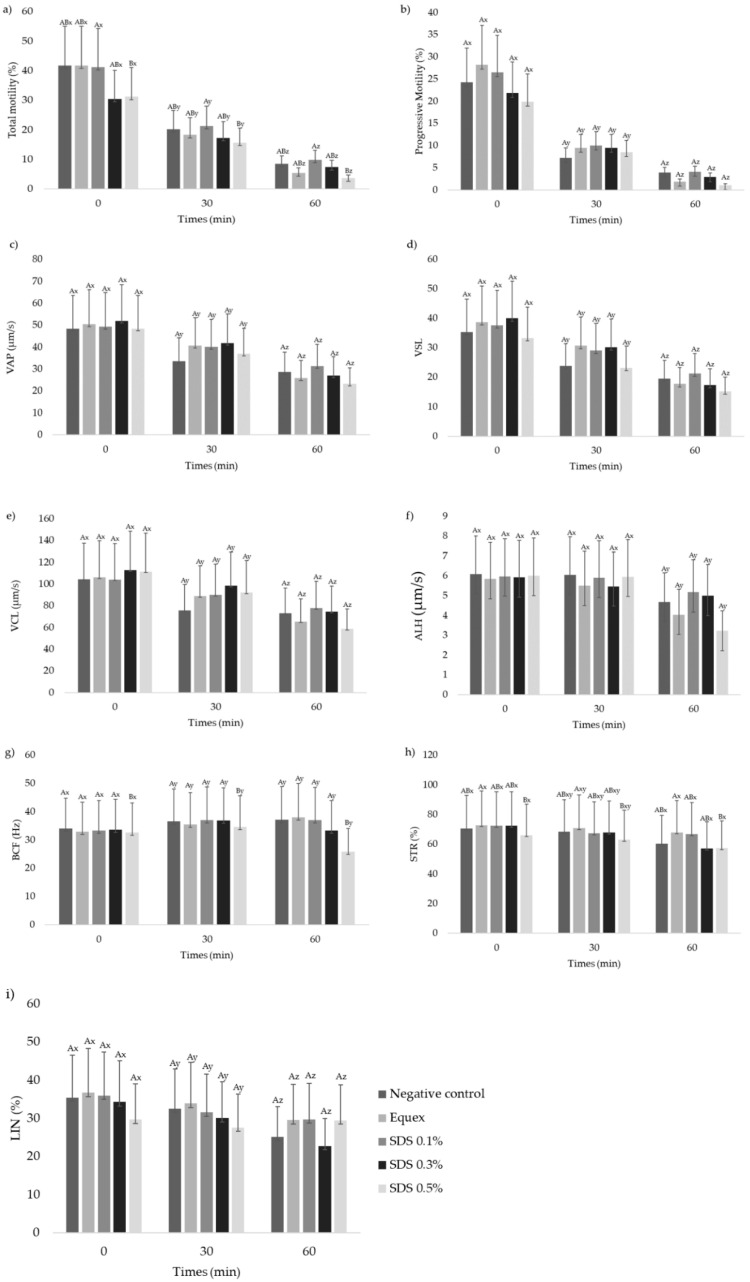
Values (mean ± SE) for the kinetic parameters (**a**–**i**) of the frozen–thawed semen of collared peccaries diluted in Tris–egg yolk, with or without detergent (Equex STM^®^ paste 0.5%; or sodium dodecyl sulfate—SDS: 0.1%, 0.3% or 0.5%) (n = 10). ^AB^ Uppercase superscript letters indicate a significant difference between experimental groups within the same time (*p* < 0.05); ^xyz^ lowercase superscript letters indicate a significant difference within the experimental group over time (*p* < 0.05).

**Figure 2 animals-13-00451-f002:**
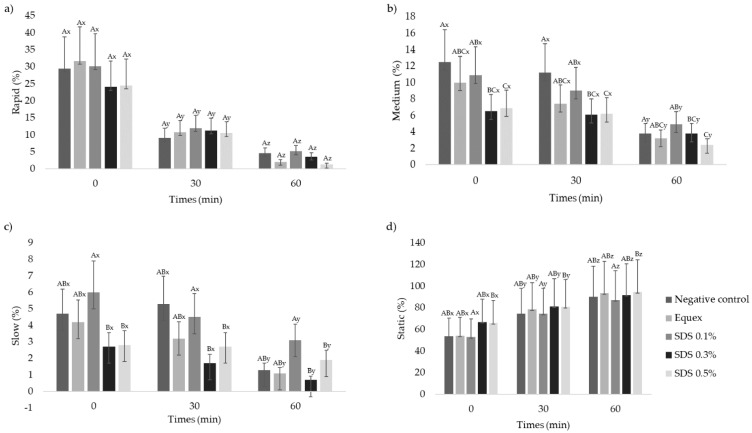
Values (mean ± SE) for sperm subpopulations rapid% (**a**), medium% (**b**), slow% (**c**) and static% (**d**) of the frozen–thawed semen of collared peccaries diluted in Tris–egg yolk, with or without detergent (Equex STM^®^ paste 0.5%; or sodium dodecyl sulfate—SDS: 0.1%, 0.3% or 0.5%) (n = 10). ^ABC^ Uppercase superscript letters indicate a significant difference between experimental groups within the same time (*p* < 0.05); ^xyz^ lowercase superscript letters indicate a significant difference within the experimental group over time (*p* < 0.05).

**Figure 3 animals-13-00451-f003:**
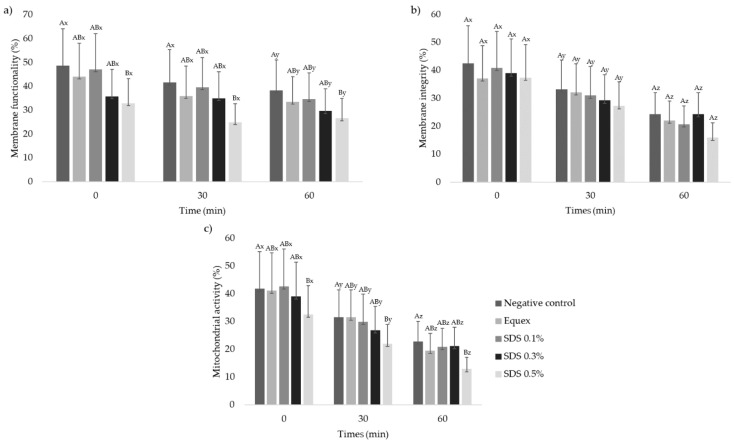
Values (mean ± SE) for membrane functionality (**a**), integrity (**b**) and mitochondrial activity (**c**) of collared peccary frozen–thawed semen diluted in Tris–egg yolk, with or without detergent (Equex STM^®^ paste 0.5%; or sodium dodecyl sulfate—SDS: 0.1%, 0.3% or 0.5%) (n = 10). ^AB^ Uppercase superscript letters indicate a significant difference between experimental groups within the same time (*p* < 0.05); ^xyz^ lowercase superscript letters indicate a significant difference within the experimental group over time (*p* < 0.05).

**Table 1 animals-13-00451-t001:** Kinetic parameter values (mean ± SE) of the collared peccaries (n = 10) fresh ejaculates used in the experiment.

	Sperm Kinetic Motility Patterns									
Treatments	Time	Total Motility (%)	Progressive Motility (%)	VAP (µm/s)	VSL (µm/s)	VCL (µm/s)	ALH (µm/s)	BCF (Hz)	STR (%)	LIN (%)
Fresh		87.8 ± 4.4	71.2 ± 4.6	57.4 ± 5.1	45.0 ± 4.4	110.0 ± 8.8	5.6 ± 0.3	33.4 ± 1.0	76.1 ± 1.1	41.3 ± 1.7

**Table 2 animals-13-00451-t002:** Mean (±SE) values and range (min–max) for number of sperm bound to the hen egg yolk perivitelline membrane in collared peccary frozen–thawed semen diluted in Tris–egg yolk, with or without detergent (Equex STM^®^ paste 0.5%; or sodium dodecyl sulfate–SDS: 0.1%, 0.3% or 0.5%) (n = 10).

Number of Bound Sperm	Frozen–Thawed Semen				
	Negative Control	Equex	SDS 0.1%	SDS 0.3%	SDS 0.5%
Mean ± SE	138.1 ± 10.8 ^A^	126.1 ± 9.0 ^A^	131.8 ± 8.0 ^A^	144.3 ± 13.6 ^A^	131.6 ± 9.3 ^A^
Min	84.4	74.3	110.0	103.4	81.0
Max	188.7	174.7	193.6	147.0	180.2

^A^ Uppercase superscript letters indicate a significant difference between the experimental groups (*p* < 0.05).

## Data Availability

The data presented in this study are available on request from the corresponding author.

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
