# Peer review of "Effect of Detergents Based on Sodium Dodecyl Sulfate on Functional Metrics of Frozen–Thawed Collared Peccary (Pecari tajacu) Semen"

_animals, 2023, doi:10.3390/ani13030451_

Round 1

Reviewer 1 Report (Previous Reviewer 2)

Even though the Authors have presented the data of the fresh semen separately, the presentations of the post-thaw (PT) semen data in line graphs are unclear. For examples, the uppercase superscript letters indicate  significant differences between experimental groups within the same time and are placed only at the control group?, and the Reviewer cannot understand the explanation given for the use of the asterisks. It is unclear when using the upper class letters (A,B), and the asterisks, which should be used for pairwise comparison.

Suggestions:

1. If used graph lines for the presentation of the PT data, i) place different letters (a,b, or c) for comparison among the time periods (0, 30 and 60 min) within the experimental groups, that is, horizontally, and ii) for comparison among the experimental groups within the same time period (for example, at 30 min for control, 0.1%, 0.3% and 0.5% SDS), then use different letters, such as x,y, or z, that is, vertically.

2. Give a brief description of the ANOVA results in the text whether treatment or storage time had any significant effect of the PT semen parameter.

3. Provide the full meaning for "LVC" and "MVC" (p. 4).

Author Response

#Reviewer 01

Even though the Authors have presented the data of the fresh semen separately, the presentations of the post-thaw (PT) semen data in line graphs are unclear. For examples, the uppercase superscript letters indicate  significant differences between experimental groups within the same time and are placed only at the control group?, and the Reviewer cannot understand the explanation given for the use of the asterisks. It is unclear when using the upper class letters (A,B), and the asterisks, which should be used for pairwise comparison.

Suggestions:

  1. If used graph lines for the presentation of the PT data, i) place different letters (a,b, or c) for comparison among the time periods (0, 30 and 60 min) within the experimental groups, that is, horizontally, and ii) for comparison among the experimental groups within the same time period (for example, at 30 min for control, 0.1%, 0.3% and 0.5% SDS), then use different letters, such as x,y, or z, that is, vertically.

ANSWER: Since the lines would be very close and it would be very difficult to allocate numerous letters indicating statistical differences, we chose to remake all the graphs and show them in the form of bars. In these, capital letters (A,B,C) indicate differences among treatments within the same time, and lowercase letters (x,y,z) indicate differences within the same treatment over time. We hope this format can be more understandable. In any case, the main differences found by the ANOVA test are highlighted in the text in the section referring to the results.

  1. Give a brief description of the ANOVA results in the text whether treatment or storage time had any significant effect of the PT semen parameter.

ANSWER: The main differences found by the ANOVA in relation to the evaluated sperm parameters are highlighted in the form of text in the results section. We emphasize that only the parameters where significant differences were found were highlighted in the text.

  1. Provide the full meaning for "LVC" and "MVC" (p. 4).

ANSWER: We have included the meaning in the text: low VAP cutoff (LVC) and medium VAP cutoff (MVC)

Reviewer 2 Report (New Reviewer)

Page 3, 2.2, 7th line: add v/v. Change to … concentrations of 0.1%, 0.3% or 0.5% v/v,…

Page 9, legend of Table 2. Change “ab” to capitals AB, as they are written into the Table.

Discussion, page 9, Last 3 lines of the 1st paragraph: “From an experimental point of view, the possibility of using isolated SDS in the formulation of extenders for semen is an advantage over the commercial detergent Equex STM Paste, since its compo-sition is not exactly known.”

More experiments including in vitro fertilization and/or in vivo artificial insemination with SDS in comparison with Equex STM Paste treated semen, were needed to support this theory. It leads in misunderstandings. The use of tested concentration of SDS is maybe more standardized process than the use of Equex of unknown composition, but who knows which of them has better experimental or fertility results?  I propose to delete these sentences or change them according to the “more standardized process”.

Author Response

#Reviewer 2

Page 3, 2.2, 7th line: add v/v. Change to … concentrations of 0.1%, 0.3% or 0.5% v/v,…

ANSWER: OK

Page 9, legend of Table 2. Change “ab” to capitals AB, as they are written into the Table.

ANSWER: OK

Discussion, page 9, Last 3 lines of the 1st paragraph: “From an experimental point of view, the possibility of using isolated SDS in the formulation of extenders for semen is an advantage over the commercial detergent Equex STM Paste, since its compo-sition is not exactly known.”

ANSWER: OK

More experiments including in vitro fertilization and/or in vivo artificial insemination with SDS in comparison with Equex STM Paste treated semen, were needed to support this theory. It leads in misunderstandings. The use of tested concentration of SDS is maybe more standardized process than the use of Equex of unknown composition, but who knows which of them has better experimental or fertility results?  I propose to delete these sentences or change them according to the “more standardized process”.

ANSWER: At the reviewer's suggestion, we added the following excerpt: "Moreover, we believe that the use of SDS alone would be a more standardized process than the addition of Equex paste, as its composition is not completely known." But we chose to keep the original sentences. We believe that the addition of this excerpt corresponds to the reviewer's expectation.

This manuscript is a resubmission of an earlier submission. The following is a list of the peer review reports and author responses from that submission.

Round 1

Reviewer 1 Report

Since the lines of the manuscript are not numbered, I would try to provide the comments in a comprehensive way for the authors, like page, title of section and/or paragraph.

2.4. Semen Freezing-Thawing Procedures

Why the callibration time before freezing is too short (40 min at 15o C and 30 min at 5 min)? Is there any reference for that protocol?

2.6. Sperm Membrane Functionality

Is the osmolarity of the hypo-osmotic solution zero? For boar semen, it is usually used a solution of 150 mOsm/L. Did the authors perform any pre-trial about it? If there is a reference, please include it.

2.6. Plasma Membrane Integrity and Mitochondrial Activity

L6: ...Halogen Lamp Set. Leica. Kista, Sweden), those cells with a head marked...

2.7. Sperm binding ability assay

The reference number 19 does not correspond to the meaning of the first sentence in that paragraph. Please, correct and provide the correct one.

3.1. Fresh Semen Parameters

...with 86.0 ± 3.3% with functional membrane, 87.6 ± 2.2 A % viable ...(Table 2).

The values of the above-mentioned variables are not the same like in Table 3 (instead of Table 2). Please, correct.

Page 6, 1st paragraph: Immediately after thawing, there was a significant reduction in the proportion of the fast sperm subpopulation concomitant with an increase in static sperm in all treat-ments (P < 0.05), as seen in Table 2. Regarding the thermal resistance test, the parameters ALH, STR, and medium and slow subpopulations remained constant up to 30 min after thawing regardless the treatment used (Table 1 and 2). However, groups containing SDS were the only ones able to maintain the VCL parameter (mm/s) similar to fresh samples during all the thermal resistance test (Tables 1).

3.3. Sperm Membrane Functionality and Integrity, and Mitochondrial Activity

...A significantly more pronounced decline for membrane functionality and mitochondrial ac-tivity was observed using 0.5% SDS over time (P < 0.5) compared to control group.

Discussion, page 8, 3rd paragraph: This argument is not so clear. Certainly, thermal stress is a well accepted cause for sperm membrane damage. However, the osmotic  and oxidative stress are, also, involved and it could be a more complete statement if they were included.

Discussion, page 9, 4th paragraph: "If, on the one hand, there is evidence of a beneficial effect of diluent containing SDS...semen [32]".

Since there is no difference between the positive control group and the SDS groups (regarding the binding test), it could not be so accurate to discuss a beneficial effect. So it is suggested: "Base on the data obtained of our results, there is no toxic effect of SDS on sperm binding phenomena. However, the assay limitations are evident and only in vivo insemination tests could reveal the real fertilizing capacity of frozen/thawed semen samples in the absence or presence of SDS, as recently postulated for canine semen [32]."

Author Response

#Reviewer 01:

Since the lines of the manuscript are not numbered, I would try to provide the comments in a comprehensive way for the authors, like page, title of section and/or paragraph.

ANSWER: Thank you for the suggestions in our manuscript, which will certainly improve its quality. We followed the template provided by the Animals journal, which does not include line numbers. We are very sorry about that.

2.4. Semen Freezing-Thawing Procedures

Why the callibration time before freezing is too short (40 min at 15o C and 30 min at 5 min)? Is there any reference for that protocol?

ANSWER: We used a protocol based on a fast-freezing curve previously established for the species, according to the following reference, which was included in the text: Silva MA, Peixoto GC, Castelo TS, Lima GL, Silva AM, Oliveira MF, Silva AR. Cryopreservation of collared peccary (Pecari tajacu) semen using different freezing curves, straw sizes, and thawing rates. Cryobiology. 2013 Aug;67(1):50-5. doi: 10.1016/j.cryobiol.2013.04.009.

2.6. Sperm Membrane Functionality

Is the osmolarity of the hypo-osmotic solution zero? For boar semen, it is usually used a solution of 150 mOsm/L. Did the authors perform any pre-trial about it? If there is a reference, please include it.

ANSWER: Despite swine is the domestic species most closely related to the peccaries, they belong to different families and there are marked physiological differences among species, including related to some reproductive aspects. For the hypoosmotic swelling test in collared peccary, we used distilled water as a hypoosmotic solutions according to what was previous determined by: Santos, E. A. A.; Sousa, P. C.; Peixoto, G. C. X.; Simao, B. R.; Oliveira, M. F.; Silva, A. R. Establishing the hypoosmotic swelling test for sperm analysis in collared peccaries (Pecari tajacu). Arq. Bras. Med. Vet. Zootec. 2013, 65, 1257–1260.  https://doi.org/10.1590/s0102-09352013000400044

2.6. Plasma Membrane Integrity and Mitochondrial Activity

L6: ...Halogen Lamp Set. Leica. Kista, Sweden), those cells with a head marked...

ANSWER: Revised accordingly.

2.7. Sperm binding ability assay

The reference number 19 does not correspond to the meaning of the first sentence in that paragraph. Please, correct and provide the correct one.

ANSWER: Sorry for the mistake, references were wrongly numbered, but they are correct at the present version. The correct reference for this statement is: Campos, L.B.; Peixoto, G.C.X.; Silva, A.M.; Souza, A.L.P.; Castelo, T.S.; Maia, K.M.; Pereira, A.F.; Silva, A.R. Estimating the bin-ding ability of collared peccary (Pecari tajacu Linnaeus, 1758) sperm using heterologous substrates. Theriogenology 2017, 92, 57–62. http://dx.doi.org/10.1016/j.theriogenology.2017.01.008

3.1. Fresh Semen Parameters

...with 86.0 ± 3.3% with functional membrane, 87.6 ± 2.2 A % viable ...(Table 2).

The values of the above-mentioned variables are not the same like in Table 3 (instead of Table 2). Please, correct.

ANSWER: Thank you for pointing this error. We revised the data according to the Table 3.

Page 6, 1st paragraph: Immediately after thawing, there was a significant reduction in the proportion of the fast sperm subpopulation concomitant with an increase in static sperm in all treat-ments (P < 0.05), as seen in Table 2. Regarding the thermal resistance test, the parameters ALH, STR, and medium and slow subpopulations remained constant up to 30 min after thawing regardless the treatment used (Table 1 and 2). However, groups containing SDS were the only ones able to maintain the VCL parameter (mm/s) similar to fresh samples during all the thermal resistance test (Tables 1).

ANSWER: Revised accordingly.

3.3. Sperm Membrane Functionality and Integrity, and Mitochondrial Activity

...A significantly more pronounced decline for membrane functionality and mitochondrial ac-tivity was observed using 0.5% SDS over time (P < 0.5) compared to control group.

ANSWER: Revised accordingly.

Discussion, page 8, 3rd paragraph: This argument is not so clear. Certainly, thermal stress is a well accepted cause for sperm membrane damage. However, the osmotic and oxidative stress are, also, involved and it could be a more complete statement if they were included.

ANSWER: Thanks for the suggestion, we included a statement related to the other stress that could damage sperm plasma membrane as the osmotic and oxidative stress.

Discussion, page 9, 4th paragraph: "If, on the one hand, there is evidence of a beneficial effect of diluent containing SDS...semen [32]".

Since there is no difference between the positive control group and the SDS groups (regarding the binding test), it could not be so accurate to discuss a beneficial effect. So it is suggested: "Base on the data obtained of our results, there is no toxic effect of SDS on sperm binding phenomena. However, the assay limitations are evident and only in vivo insemination tests could reveal the real fertilizing capacity of frozen/thawed semen samples in the absence or presence of SDS, as recently postulated for canine semen [32]."

ANSWER: Revised accordingly. Thank you.

Reviewer 2 Report

In this study attempts were made to investigate the effects of Equex STM and different concentrations of sodium dodecyl sulfate (SDS) on post-thaw quality of collared peccary (Pecari tajacu) semen. This manuscript suffers significantly from poor organization and presentation of the data that reduces its scientific relevance. The Reviewer suggests that the following comments would be helpful to improve the quality of the manuscript.

Comments

1. Objective

Missing is Equex STM. What is the composition of Equex, and the concentration of SDS? Such information should be included in the Discussion.

2. M & M

a) Re-check the CASA setting and analyzed parameters (appropriate citation?).

b) Should consider to use "motility characteristics and motion parameters" or "motility characteristics and kinematic parameters".

c) The units of the velocity parameters -micrometers/s (mm/s), and ALM (mm). Define "MVV" and "LVV"

d) Missing is LIN (VSL/VCL × 100) in Table 1.

e) Should consider to re-run the statistical analysis and re-write the results. The parameters of the fresh semen (from Tables 1-4) should not be included in the statistical analysis and should be presented separately in another table. There are 2 main factors, treatment (untreated -control and treated) and storage time (3) giving a 2 × 3 factorial design (2-way ANOVA followed by the Tukey’s post hoc test). Re-run the analysis simultaneously for the 12 measurements of each parameter (Tables 1-3), and use only one significant level  (for examples, a,b,c - different letters in the same column are significant) because the data shown the tables are too confusing.

f) Present an ANOVA table with the main factors and their interaction.

3. Results and Discussion

A re-run of the statistical analysis will probably give different results and their interpretation.

4. Delete Ref. #35.

Author Response

#Reviewer 2:

In this study attempts were made to investigate the effects of Equex STM and different concentrations of sodium dodecyl sulfate (SDS) on post-thaw quality of collared peccary (Pecari tajacu) semen. This manuscript suffers significantly from poor organization and presentation of the data that reduces its scientific relevance. The Reviewer suggests that the following comments would be helpful to improve the quality of the manuscript.

Comments

  1. Objective

Missing is Equex STM.

ANSWER: In the objective section, we rewrote it for: “…the aim of the present study was to verify the effect of including Equex STM® Paste or different concentrations of sodium dodecyl sulfate (SDS) on various functional metrics of frozen-thawed peccary semen…”

What is the composition of Equex, and the concentration of SDS? Such information should be included in the Discussion.

ANSWER: Since the absence of information related to the detailed composition of Equex is one of the justificative for the execution of our study, this information is stated at the second paragraph of introduction as “…Furthermore, the exact composition of the paste in terms of excipients and other compounds is not known, even if manufacturers highlight the sodium dodecyl sulfate (SDS) as the main active component albeit at an unknown concentration…”

Anyway, we also included some information at the first paragraph of discussion as suggested: “…From an experimental point of view, the possibility of using isolated SDS in the for-mulation of extenders for semen is an advantage over the commercial detergent Equex STM Paste, since its composition is not exactly known.”

  1. M & M
  2. a) Re-check the CASA setting and analyzed parameters (appropriate citation?).

ANSWER: We revised it and included the settings that were previously stablished by: Souza, A.L.P.; Lima, G.L.; Peixoto, G.C.X.; Silva, A.M.; Oliveira, M.F.; Silva, A.R. Use of Aloe vera-based extender for chilling and freezing collared peccary (Pecari tajacu) semen. Theriogenology 2016, 85, 1432–1438. https://doi.org/10.1016/j.theriogenology.2016.01.007.

  1. b) Should consider to use "motility characteristics and motion parameters" or "motility characteristics and kinematic parameters".

ANSWER: We substitute it according to reviewer’s suggestion.

  1. c) The units of the velocity parameters -micrometers/s (mm/s), and ALM (mm).

ANSWER: So sorry for this immense typing mistake. We revise it in all the text and result tables.

Define "MVV" and "LVV"

ANSWER: We revised it for LVC and MVC that means Low VAP Cutoff and Medium VAP Cutoff, respectively. Thank you for pointing this mistake.

  1. d) Missing is LIN (VSL/VCL × 100) in Table 1.

ANSWER: Sorry for the mistake. We included it in the table in this new version.

  1. e) Should consider to re-run the statistical analysis and re-write the results. The parameters of the fresh semen (from Tables 1-4) should not be included in the statistical analysis and should be presented separately in another table.

ANSWER: In the statistical analysis section, we clarify that, initially, fresh semen data were compared to the frozen-thawed-others using Dunnet's test. And this strategy was important, because we were able to detect that 0.1% SDS would be able to maintain some parameters (such as ALH, BCF and VCL) at values similar to those of fresh semen. This comparison with fresh semen is indicated in the tables by means of an asterisk and explained in the captions. We believe that this is the best way to show the data, because if we separate them into different tables, we would not be able to show these nuances in which diluents manage to keep values similar to fresh samples.

e.2) There are 2 main factors, treatment (untreated -control and treated) and storage time (3) giving a 2 × 3 factorial design (2-way ANOVA followed by the Tukey’s post hoc test). Re-run the analysis simultaneously for the 12 measurements of each parameter (Tables 1-3), and use only one significant level (for examples, a,b,c - different letters in the same column are significant) because the data shown the tables are too confusing.

ANSWER: As also explained in the statistical analysis section, we compared only the frozen-thawed groups (excluding fresh samples) using two-way ANOVA followed by Tukey's test. In this analysis, we considered both the variables related to the different treatments (differences expressed by uppercase letters - ABC) and time (differences expressed by lowercase letters - abc). We understand that there is a lot of information in the same table, but we tried to explain it in the captions. If we were to express different tables for each variable, instead of four tables, we would have 12 tables in all, and we really don't believe this would be feasible for publication. Therefore, we ask reviewer to consider maintaining the data presentation at the present form.

  1. f) Present an ANOVA table with the main factors and their interaction.

ANSWER: Please view the previous answers.

  1. Results and Discussion

A re-run of the statistical analysis will probably give different results and their interpretation.

ANSWER: As previously explained and shown in the methodology section referring to statistical analysis, the suggestion made by the reviewer had already been conducted and the data presented here already follow the requested premise.

  1. Delete Ref. #35.

ANSWER: We revised all the references in the entire document.

Reviewer 3 Report

In this study, Moreira et al. analyze the potential of SDS as a cryosupplement in collared peccary sperm preservation. This is an interesting animal model to begin with, and even more so SDS, which is traditionally more known in research as a detergent. While SDS has been studied previously, the study addresses gaps that are still missing to ellucidate the behavior of SDS on the sperm structure and function. The study is well designed, and the collected data are properly discussed. I also appreciate a good flow of the manuscript that reads well.

I have several questions most of which stem from my own curiosity:

- Whilst this was not the aim of the study, oxidative stress is known to play an important role in sperm cryodamage. In their opinion, does SDS exhibit any antioxidant properties or does it just indirectly stabilize the membranes to make them more resilient towards oxdidative damage?

- Obviously, higher SDS concentrations may have detrimental effects on the sperm integrity and vitality. How did the authors decide on the final SDS concentration range used in the experiments?

- Could the authors provide the manufacturer of the optical microscope used for the HOS test?

- The authors could briefly summarize any limitations the study could suffer from?

Author Response

#Reviewer 03:

In this study, Moreira et al. analyze the potential of SDS as a cryosupplement in collared peccary sperm preservation. This is an interesting animal model to begin with, and even more so SDS, which is traditionally more known in research as a detergent. While SDS has been studied previously, the study addresses gaps that are still missing to ellucidate the behavior of SDS on the sperm structure and function. The study is well designed, and the collected data are properly discussed. I also appreciate a good flow of the manuscript that reads well.

ANSWER: Thank you for the words.

I have several questions most of which stem from my own curiosity:

- Whilst this was not the aim of the study, oxidative stress is known to play an important role in sperm cryodamage. In their opinion, does SDS exhibit any antioxidant properties or does it just indirectly stabilize the membranes to make them more resilient towards oxidative damage?

ANSWER: In dogs, it was recently demonstrated that the addition of the commercial detergent Equex STM (whose main active ingredient is SDS) to the semen extender promotes a reduction in the amount of reactive species, contributing to the reduction of oxidative stress resulting from the freezing and thawing process (Schafer-Somi et al., Cryobiology 100 (2021) 63–71). However, the effect of SDS seems to be species specific and we would need a further study to prove the existence of some antioxidant action of SDS in other groups of species, such as peccaries.

- Obviously, higher SDS concentrations may have detrimental effects on the sperm integrity and vitality. How did the authors decide on the final SDS concentration range used in the experiments?

ANSWER: In the description of the experimental design, in the Materials and Methods section, we describe that the SDS concentrations were established from previous studies carried out in pigs, which are the domestic species most phylogenetically related to peccaries.(Silva, M.H.M.; Valadão, L.; Silva, F.M. Effect of the tensioactive Agent SDS on extender’s boar semen viability after thawing. Am. J. App. Sci. 2021, 18, 9–14. https://doi.org/10.3844/ajassp.2021.9.14)

- Could the authors provide the manufacturer of the optical microscope used for the HOS test?

ANSWER: A phase contrast microscope from the company Alttion (Wuzhou City, Guangxi Province, China) was used. We added this information to the text.

- The authors could briefly summarize any limitations the study could suffer from?

ANSWER: We believe that the first limitation that actually prompted the development of the work was the unknown composition of the commercial detergent Equex, which encouraged us to test the different concentrations of SDS. In addition, as we work with wild animals, there are ethical limitations regarding the use of animals as care with containment and the use of the minimum sample number possible.

Round 2

Reviewer 2 Report

The Authors have addressed most of my comments except the below-mentioned comments.

a) Should consider to re-run the statistical analysis and re-write the results. The parameters of the fresh semen (from Tables 1-4) should not be included in the statistical analysis and should be presented separately in another table. There are 2 main factors, treatment (untreated -control and treated) and storage time (3) giving a 2 × 3 factorial design (2-way ANOVA followed by the Tukey’s post hoc test). Re-run the analysis simultaneously for the 12 measurements of each parameter (Tables 1-3), and use only one significant level  (for examples, a,b,c - different letters in the same column are significant) because the data shown the tables are too confusing.

b) Present an ANOVA table with the main factors and their interaction.

1)The Review still insists that the parameters for the fresh semen should not be included in the statistical analysis. They should be placed in another table (one table) or remained in the same table. When compared the fresh semen with the frozen-thawed (FT) semen, you are 100% sure to get a significant difference. Boar semen when frozen always gives reduced parameters (motility, plasma membrane integrity, etc.). There is no logical reason for comparing the fresh semen with the FT semen. It only just "masked" the effects of SDS on the post-thaw semen. Furthermore, I do not agree with the Authors, suggesting that they will be 12 tables, a bit of exaggeration. It will be about 6 tables, providing that the main findings of the interactions are given in the text.

2)The experimental design shows that there is a 2 × 3 factorial design, without the fresh semen.

Author Response

ANSWER: We accepted all the reviewer suggestions. Therefore, we reorganized the tables, providing results related to fresh semen in a separate table. Some of the fresh semen characteristics were only described in the text. Moreover, we rewrote the other tables to show only the ANOVA results, thus considering a 2 x 3 factorial design. Based on these statements, we excluded all the discussions related to the comparisons between fresh and frozen samples. Discussions are now only focused on the effects of SDS-based extenders and time. In general, the exclusion of the comparison with the fresh samples did not provoke great changes in the discussion, since we had already done the two-way ANOVA followed by the Tukey test as requested by the reviewer.